# Making Patient-Specific Treatment Decisions Using Prognostic Variables and Utilities of Clinical Outcomes

**DOI:** 10.3390/cancers13112741

**Published:** 2021-06-01

**Authors:** Pavlos Msaouel, Juhee Lee, Peter F. Thall

**Affiliations:** 1Department of Genitourinary Medical Oncology, The University of Texas MD Anderson Cancer Center, Houston, TX 77030, USA; 2Department of Translational Molecular Pathology, Division of Pathology and Laboratory Medicine, The University of Texas MD Anderson Cancer Center, Houston, TX 77030, USA; 3Department of Statistics, University of California Santa Cruz, Santa Cruz, CA 95064, USA; juheelee@soe.ucsc.edu; 4Department of Biostatistics, Division of Basic Sciences, The University of Texas MD Anderson Cancer Center, Houston, TX 70030, USA; rex@mdanderson.org

**Keywords:** individualized inferences, patient-specific decision-making, precision medicine, prognostic biomarkers, utilities

## Abstract

**Simple Summary:**

Clinicians often erroneously discount prognostic information as unlikely to change patient management. This is fueled by the mistaken belief that only “predictive” subgroups or biomarkers can modify the differences in clinical benefit between treatment choices. We use the treatment of metastatic clear cell carcinoma as an example to illustrate how clinical decisions can be informed by prognostic variables. Diametrically opposite decisions can be made depending on individual patient prognosis and on the clinical outcome of interest that clinicians choose to focus on. We also demonstrate why such patient-specific treatment decisions inevitably should be guided by each patient’s goals and values, which can be explicitly represented by utility functions.

**Abstract:**

We argue that well-informed patient-specific decision-making may be carried out as three consecutive tasks: (1) estimating key parameters of a statistical model, (2) using prognostic information to convert these parameters into clinically interpretable values, and (3) specifying joint utility functions to quantify risk–benefit trade-offs between clinical outcomes. Using the management of metastatic clear cell renal cell carcinoma as our motivating example, we explain the role of prognostic covariates that characterize between-patient heterogeneity in clinical outcomes. We show that explicitly specifying the joint utility of clinical outcomes provides a coherent basis for patient-specific decision-making.

## 1. Introduction

Although clinicians regularly make individualized decisions for their patients, the question of how statistical evidence can be used most efficiently to guide such decisions has remained elusive [1]. Due to the abundance of approved therapeutic options and the availability of large observational datasets [2,3], we have used the management of metastatic clear cell renal cell carcinoma (mccRCC), the most common kidney cancer subtype, as our motivating example to illustrate a utility-based approach to patient-specific decision-making. The International Metastatic Renal Cell Carcinoma Database Consortium (IMDC) has facilitated the development of robust prognostic scores for mccRCC [3], which are used by organizations such as the National Comprehensive Cancer Network (NCCN) to guide treatment selection [4]. The IMDC risk score for mccRCC is calculated for individual patients by combining biomarkers and clinical variables such as anemia, thrombocytosis, neutrophilia, hypercalcemia, Karnofsky performance status, and time from diagnosis to treatment [3]. Based on their individual IMDC scores, patients can be classified as having favorable-, intermediate-, or poor-risk disease, reflecting the substantial heterogeneity in survival outcomes among patients with mccRCC [3,5,6]. The IMDC score thus is a very useful tool to examine how individual covariates can be used to inform clinical decisions.

In this article, we have used the mccRCC and IMDC paradigms to illustrate the statistical and patient-specific considerations involved in making individualized clinical decisions. In the examples given below, for simplicity, we will choose between two treatment options for each mccRCC patient considered. The concepts are generalizable to more complex treatment options and other scenarios, such as diagnostic and surveillance strategies. The sources of our illustrative data will be randomized clinical trials (RCTs), because they are simple but powerful clinical experiments that can provide unbiased effect estimates [7,8]. This will allow us to focus on how clinicians can make patient-specific decisions informed by statistical analyses of data, which may be derived from randomized or non-randomized sources. We argue that coherent individualized treatment decisions may be carried out as three consecutive tasks (Table 1): (1) estimating key parameters of a statistical model, (2) using prognostic information to convert these parameters into clinically interpretable outcomes, and (3) specifying joint utility functions to quantify risk–benefit trade-offs between clinical outcomes.

## 2. Statistical Estimation

The primary goal of statistical models and methods is to provide accurate and precise estimates of conceptual objects termed “parameters”, such as the hazard rate, probability of response, median survival time, or the effects of patient prognostic variables on a clinical outcome. For comparing treatments, the parameters most commonly estimated from RCTs in oncology and other medical fields are ratios such as the hazard ratio (HR) or the odds ratio (OR) [9,10,11]. Ratios express the relative effects of one treatment compared with another, and obtaining unbiased estimates of comparative treatment effects represented by ratios is the principal goal of RCTs.

HRs are ratios of hazard rates, and estimated HRs are used to compare event–time distributions, with survival time being the most common primary endpoint in oncology. The hazard rate at a particular time from the start of therapy is the instantaneous rate of a patient experiencing a particular outcome, such as death in an overall survival analysis or failure (death or disease progression) in a progression-free survival analysis, given that the patient has not yet experienced the event. If the hazard rate does not vary with time, then it is expressed in a single value, h. The odds of an event are the probability, P, that it will occur divided by the probability, 1 − P, that it will not, formally P/(1 − P). For HRs and ORs, conventionally, the treatment group (T) is used in the numerator and the control group (C) in the denominator, so the HR = h_T_/h_C_ and the OR = (P_T_/(1 − P_T_))/(P_C_/(1 − P_C_)). Thus, HR = 1.0 means that the treatment and control groups have the same hazard rate of the event of interest. HR < 1.0 indicates that the treatment group has a lower hazard rate than that of the control group, whereas the opposite is true for HR > 1.0. It is a fundamental mistake to interpret a HR as a measure of relative risk; hazard rates are instantaneous rates that can range from zero to infinity, whereas risks are probabilities that range from zero to one [11]. Probabilities can also be expressed equivalently as percentages, instead of proportions, in which case they can range from 0% to 100%. Like hazard rates, the treatment and control group odds, P_T_/(1 − P_T_) and P_C_/(1 − P_C_), used to compute the OR, can each range from zero to infinity.

The clinical meaningfulness of effect size differences is a key concept in interpreting HRs and ORs. For survival outcomes, it is commonly accepted in practice that HRs ranging between 0.8 and 1.25 (or less conservatively between 0.9 and 1.1) represent clinically non-meaningful differences. HRs lower than 0.8 or higher than 1.25 may be considered to be clinically meaningful effect size differences, favoring the treatment or control group, respectively (Figure 1). The uncertainty of the effect size estimate is commonly represented by the width of a 0.95 confidence interval (CI). As illustrated in Figure 1 (middle estimate), effect size estimates with wide CIs can be inconclusive due to being statistically compatible with both clinically meaningful and non-meaningful differences. Two treatments can be considered practically equal when the effect size estimate is only compatible with clinically non-meaningful differences (Figure 1, bottom estimate). Accurately interpreting CI estimates can be a complex endeavor, and frequentist CIs (the most commonly presented CIs) can counterintuitively be narrow and imprecise if, for example, the underlying statistical model does not fit the clinical data well [12]. For the purposes of the present article, we make the common assumption that narrow CIs indicate increased precision and refer those interested to detailed overviews of frequentist CIs and Bayesian credible intervals for further discussion of the related nuances [12,13,14].

### 2.1. Advantages and Limitations of Estimated Parameters

Because hazard rates and odds take values between zero and infinity, HRs and ORs have the mathematical advantage that they can easily accommodate inverting the reference group: a HR of 0.5 with the control group as the denominator is equivalent to an inverted HR of 1.0/0.5 = 2.0 with the treatment group as the denominator. This symmetry does not hold for a risk ratio, RR = P_T_/P_C_, which is sensitive to “framing effects” because probabilities have an upper limit of 1. If the RR = 2.0 and the denominator for the control group has a risk of P_C_ = 0.6, then the numerator for the treatment group would be P_T_ = 1.2, which violates the axiom that probabilities cannot have values greater than 1.0. If the RR = 2.0, this implies that P_C_ can be at most 0.5. This sort of numerical problem is a major reason why HRs and ORs are the parameters that are predominantly estimated in order to compare treatments. Typically, model-based estimates are computed as the natural logarithm of the HR (log_e_ HR) or OR (log_e_ OR) on a mathematical scale that we will call the “estimation scale.” Exponentiation of the estimation scale parameters produces the HR or OR estimates commonly presented in the analysis of RCTs. An additional advantage of using hazard rates to model survival outcomes statistically is that they are directly identifiable from censored event time data and allow for general censoring processes [15].

Parameters directly estimated by statistical models are mathematical constructs that, on their own, may lack clinical meaning. Neither clinicians nor patients can easily understand what a hazard rate represents or how it pertains to clinical care. However, parameters defined on the estimation scale can be converted into clinically interpretable values that correspond to the distinct “outcome scales” produced during the clinical outcome prediction step (second step), which we will describe below (Table 1).

An additional consideration that has generated much discussion is the collapsibility of estimated parameters [16,17,18,19,20,21]. A parameter is collapsible when its value for the whole group (also known as the “marginal” value) is a weighted average of its values for patient subgroups (known as a “conditional” value for a subgroup of interest) and is therefore invariant when adjustment for these subgroups is performed. In contrast, non-collapsible parameters demonstrate the counterintuitive behavior whereby the marginal value is not a weighted average of the subgroup values and, therefore, adjustment for subgroups alters the measured parameter even in the absence of confounding or effect modification [17,20,21]. A numerical example is provided in Appendix A. Parameters can be non-collapsible when the outcome is categorical or represents a time-to-event, but are always collapsible when the outcome is continuous. Interested readers are referred to two detailed recent overviews describing the nuances of this phenomenon [17,18], which is pertinent to clinical inference because two of the most commonly estimated parameters in clinical research, log_e_ ORs and log_e_ HRs (and the corresponding ORs and HRs), are non-collapsible.

An intense debate is ongoing, with some considering the non-collapsibility of ORs and HRs to be an advantage for statistical modeling because the mathematical features that render a parameter non-collapsible also provide the properties that facilitate statistical analyses on the estimation scale, including the unlimited range of odds and hazard rates and the fact that they are non-linear functions of probabilities [9,22]. Others, however, argue that collapsible measures such as risk difference and mean survival time difference are preferable parameters to use because, in addition to being difficult to interpret clinically, the non-collapsibility of ORs and HRs adds to problems of transportability and makes it harder to compare estimates across studies [17,23]. As shown in Table 1, our proposed framework harmonizes these contrasting views by distinguishing between the tasks of statistical estimation on the estimation scale and clinical outcome prediction on the outcome scales. The first task can be facilitated by the mathematical advantages of non-collapsible parameters, and this is why log_e_ ORs and log_e_ HRs are so much more commonly used for statistical estimation than their collapsible counterparts. As has been argued eloquently by Sander Greenland [24], statistical models used in health sciences research act as smoothing or noise reduction devices and not as embodiments of physical laws. A key step is to utilize additional information, such as an individual patient’s baseline risk, to convert statistical estimates, such as log_e_ ORs and log_e_ HRs, into clinically interpretable collapsible outcomes such as risk differences and mean survival time (on the outcome scales).

### 2.2. Subgroup Analysis on the Estimation Scale

Clinicians commonly attempt to make patient-specific inferences by looking for differences between subgroups in estimation scale parameters, such as the log_e_ HR. This practice is commonly referred to as looking for “predictive” subgroups, and/or biomarkers corresponding to these subgroups, that can guide treatment choices for individual patients. Mathematically, these “predictive” biomarkers have a multiplicative relationship with the parameter of interest (e.g., the log_e_ HR) in the regression equation used for statistical estimation. For example, to determine the predictive effect of IMDC risk on overall survival in an RCT comparing two treatments, the statistical model should be constructed on the estimation scale to include a product or “interaction” term between treatment and IMDC risk, b_1_ · (treatment) + b_2_·IMDC + b_3_· (treatment) · IMDC, where the term (treatment) equals 1 for one treatment and equals 0 for the other, and b_1_, b_2_, and b_3_ are parameters that quantify the magnitude of each effect. This is in contrast with “prognostic” variables that only have an additive effect on the estimation scale: b_1_· (treatment) + b_2_·IMDC. The terminology “prognostic” and “predictive” can be misleading because “prognostic” biomarkers can be efficiently used to predict clinical outcome heterogeneity. However, we will adopt these terms for the purposes of the present paper, with the understanding that an additive effect on the estimation scale can powerfully impact clinical outcomes once transformed to the outcome scale (Table 1).

Predictive subgroup effects from RCT data are commonly represented by forest plots, which are sets of point estimates and confidence intervals (CIs) that were originally developed for meta-analysis of RCTs [25], but are increasingly used for subgroup analyses within single RCTs. When interpreting forest plots, such as the example shown in Figure 2, the vertical dotted line corresponding to the overall treatment effect is more important than the vertical line corresponding to the no-effect point [26]. In Figure 2, the HR estimates from all subgroups other than subgroup 5 are statistically compatible with the overall treatment effect. In actual RCT data, subgroups 1–4 (Figure 2) are by far the most commonly encountered scenarios, whereas patterns such as that of subgroup 5, in which the subgroup HR estimate is incompatible with the overall treatment effect, are exceedingly rare. For example, when looking at first-line immunotherapy combinations for mccRCC, all the forest plots of subgroups from every major publication of phase 3 RCT trials published to date correspond to one of the scenarios represented by subgroups 1–4, and thus they do not provide any conclusive evidence of subgroup differences on the estimation scale [27,28,29,30,31,32]. The authors of the STAMPEDE phase 3 RCT, which established the addition of abiraterone to androgen deprivation therapy as a treatment option for locally advanced or metastatic prostate cancer, estimated effects in many implausible subgroups before finally identifying one that more closely resembled subgroup 5 than subgroups 1–4 [33]. This practice is known, generally, as “data dredging”, and the authors of STAMPEDE used this as an example of the type I error inflation that muddies the interpretation of forest plots [33,34]. It is well known that if one looks at enough subgroups in a dataset, eventually a seemingly “statistically significant” treatment effect will emerge even if, in fact, there is no actual treatment effect at all [35].

A major reason why predictive subgroups are identified so rarely from RCT data is that these are multiplicative treatment–subgroup interactions that often are weak and thus would require very large sample sizes to be estimated accurately [36]. As a result, a fundamental assumption behind most statistical models used in clinical drug development is that predictive subgroup effects can safely be ignored, whereas prognostic information should always be incorporated to increase the efficiency of inferences from the fitted model [10]. Indeed, the statistical models used for all primary endpoint analyses for the phase 3 RCTs of first-line immunotherapy combinations in mccRCC did not incorporate any predictive subgroup effects but accounted for prognostic subgroup information, such as the IMDC risk [27,28,29,30,31,32]. As we demonstrate below, prognostic subgroup information can be used similarly to make patient-specific inferences using strategies informed by the “risk-modeling” approach described by the Predictive Approaches to Treatment effect Heterogeneity (PATH) consensus statement [37]. Although predictive biomarkers certainly do exist and can be clinically valuable when identified, it is often more practical to inform their discovery using laboratory experiments and robust translational correlative analyses. Once credible potential for predictive biomarker interactions has been established preclinically, to obviate problems with post hoc data dredging, RCT designs can focus prospectively on identifying such effects [37,38,39].

Potential confounding is another caveat that is often overlooked when examining forest plots to look for predictive subgroups/biomarkers [40]. We will employ causal diagram techniques using directed acyclic graphs (DAGs) to show how such confounding can happen even in RCT datasets [41,42,43]. Consider a hypothetical RCT to compare the use of the anti-PD1 immune checkpoint inhibitor (ICI) nivolumab plus the anti-CTLA4 ICI ipilimumab versus the tyrosine kinase inhibitor sunitinib as a first-line therapy in patients with mccRCC. The IMDC risk score is used for treatment choices and is known to affect overall survival (OS) outcomes. Suppose that correlative tissue and blood analyses from the RCT also identified a gene signature that seems to predict lower OS with nivolumab plus ipilimumab and higher OS with sunitinib in a particular patient subset. Further assume that mutations in the *PBRM1* gene, the second most commonly mutated gene in mccRCC, which is known to affect OS outcomes [44,45,46], alter the expression of the hypothetical gene signature. As shown in Figure 3A,B, although randomization removes confounding induced by the IMDC score or by any other potential confounders that can affect treatment choice, it does not remove the confounding induced by *PBRM1* mutation status or by any other putative confounders affecting the gene signature biomarker and OS. Thus, to properly estimate the mediating effect of the gene signature on OS by treatment choice we need to include knowledge of *PBRM1* mutation status in the statistical regression model. This suggests that, in general, prospectively modeling treatment interaction effects based on evidence from preclinical correlative studies is preferable to forest plots for effects in subgroups that were not specified a priori [37,38]. Notably, as shown in Figure 3B, randomization turned the IMDC risk score from a confounder into a purely prognostic biomarker that, when incorporated additively into the statistical regression model, can improve the power of null hypothesis tests [16,47]. We next explore how one can also harness the information provided by prognostic biomarkers, such as IMDC score, to predict clinical outcomes for individual patients.

## 3. Clinical Outcome Prediction

Estimated parameters, such as log_e_ HR, can be used to calculate estimates of clinically interpretable parameters, such as median or mean survival time, three-month survival probability, or one-year survival probability, which correspond to easily interpretable clinical outcomes. Knowledge of each patient’s baseline risk using prognostic variables such as IMDC score allows more accurate and individualized predictions of survival outcomes.

Instead of using a clinically opaque metric such as the HR, one might be tempted to compare two treatments using only clinical outcome parameters, such as differences in median survival times or absolute risk reduction (ARR). Denoting D as time of death, this is defined as the difference between the control and treatment probabilities of dying by a given time, such as ARR = P_C_(D < 3 months) − P_T_(D < 3 months). A remarkable paper by Snapinn and Jiang [48] shows why such comparisons can be problematic when performed in isolation. They argue that, for any survival outcome, the worse the prognosis (determined, for example, by the IMDC score in mccRCC), the greater the difference in survival at a given time point yielded by a given treatment benefit, as determined by the HR. Conversely, a worse prognosis also will reduce the impact of that same treatment benefit on the time survived, thus resulting in smaller differences in median survival. This was shown to be the case when assuming an exponential distribution for survival time, which assumes a constant hazard rate and is the most commonly used distribution when modeling survival times in medicine. The same phenomenon was also noted when the two treatment groups had the same shape parameter in the more general Weibull survival distribution, which allows for increasing or decreasing hazard rates over time [48].

To see the profound effects that prognosis and parameter domain can have on a treatment comparison, consider a hypothetical phase 3 RCT comparing a new immunotherapy drug named superlumab with the approved single-agent ICI nivolumab as salvage therapies for patients with mccRCC. Suppose that superlumab has a worse side effect profile, is more expensive, and requires longer and more frequent infusions than nivolumab, and therefore is associated with worse quality of life (QOL). We assume exponential distributions for OS times with HR = 0.5 favoring superlumab over nivolumab, irrespective of IMDC risk score. Figure 4 shows the OS curves of patients for each treatment, stratified by IMDC favorable- or poor-risk subgroups. An HR of 0.5 means that the median survival of patients treated with superlumab is double that of patients treated with nivolumab. Thus, if patients with favorable-risk disease, as determined by IMDC, have a median OS of 18 months when treated with nivolumab, then the median OS is 36 months when treated with superlumab, resulting in a difference in median survival times of 36 − 18 = 18 months. Due to their much worse overall prognosis when compared with their favorable-risk counterparts, our hypothetical patients with poor-risk disease determined by IMDC have a median OS of 2 months when treated with nivolumab and 4 months when treated with superlumab, yielding a difference in median survival times of only 4 − 2 = 2 months. Thus, when focusing on survival time as the clinical outcome of interest, for a HR of 0.5, in terms of differences between median survival times the benefit of superlumab is clearly much larger for patients with favorable-risk mccRCC. Under an exponential distribution, mean survival = 1.44 · median survival; thus, the difference between the mean survival times of superlumab and nivolumab for patients with favorable-risk IMDC is 26 months, whereas for poor-risk IMDC it is still only 2.9 months. The difference between mean survival differences, 26 − 2.9 = 23.1 months in favorable- versus poor-risk patients, is even larger due to the mean-to-median scaling parameter, 1.44. Therefore, in terms of either median or mean survival, the superiority of superlumab over nivolumab is clinically very meaningful in patients with favorable-risk mccRCC, whereas the difference is quite small for patients with poor-risk mccRCC.

The survival probability of patients treated with each drug, accounting for their IMDC score, may be calculated using the formula P(D > t) = *e*^−ht^, where *e* = 2.718 is Euler’s number, h is the hazard rate, and t is the time point of interest (e.g., 3 months). The hazard rate, h, depends on the prognosis of each patient, as expressed by their IMDC score, with poor-prognosis patients having higher hazard rates than favorable-prognosis patients. Because the HR = 0.5, patients with the same IMDC risk treated with superlumab have half the hazard rate of those treated with nivolumab. The hazard rate, h, can be calculated from the median survival time using the formula h = log_e_(2)/(median survival). Thus, for patients with an IMDC favorable risk, the three-month survival probability is P(D > 3) = 0.89 if treated with nivolumab and 0.94 if treated with superlumab, yielding an ARR of 0.05. Conversely, for patients with poor-risk IMDC scores, the three-month survival probability is P(D > 3) = 0.35 for those treated with nivolumab and 0.60 for those treated with superlumab, yielding an ARR of 0.25. Thus, when focusing on ARR at three months as the parameter of interest, the results are clearly much more impressive for patients with poor-risk mccRCC. Although ARR is a well-established, widely accepted, and clinically meaningful metric [49,50,51], the two outcome criteria (ARR versus median or mean survival differences) lead to opposite conclusions in our hypothetical RCT. ARR favors the use of superlumab more in patients with poor prognosis, whereas other clinically meaningful parameters such as median or mean survival differences favor superlumab more in patients with favorable prognosis, despite the HR being the same across prognostic subgroups. The dependence of comparative treatment effects on both the prognosis and the outcome domain is illustrated in Figure 4. The figure shows that the ARR probability difference, computed at a particular time, and the difference between median survival times are actually two very different parameters that depend on the particular shapes of the two survival curves being compared. Moreover, the shapes of the two curves may both change dramatically with the prognosis. This phenomenon applies quite generally to any clinical setting with time-to-event outcomes. For example, in clinical scenarios where adjuvant therapy is considered, patients with lymph node–positive disease are generally more likely to have a higher risk of disease recurrence compared with those with lymph node–negative disease, paralleling the higher death hazard rates seen in patients with IMDC poor-risk versus favorable-risk mccRCC.

The assumed constancy of log_e_ HRs across patient subgroups is a major reason why this parameter is the one used most commonly for survival comparisons on the estimation scale. However, practicing clinicians must make decisions for individual patients based on clinically interpretable outcomes. However, different clinical outcome measures may directly contradict each other. For example, if a new agent is associated with increased mean survival and decreased quality of life (QOL) compared with a control, it becomes inevitable that therapeutic decision-making should account for each patient’s specific goals and values in terms of the trade-off between survival time and QOL. In the following section, we show how this may be made explicit.

## 4. Making Clinical Decisions

Consider a clinician seeing two mccRCC patients, A and B, who differ only in their IMDC prognostic group, similarly to the patients treated with superlumab or nivolumab in our hypothetical phase 3 RCT (Figure 4). The primary wish of each patient is to be alive at three months from treatment initiation to see their child graduate from college. Their secondary wish is to have good QOL. The only difference between the two patients is that patient A has IMDC-favorable-risk, whereas patient B has IMDC-poor-risk mccRCC. For patient A, P(D > 3 with superlumab) = 0.94 and P(D > 3 with nivolumab) = 0.89, so she is very likely to be alive at three months, regardless of whether she receives nivolumab or superlumab. However, in terms of QOL, the improved side effect profile and logistical and financial advantages of nivolumab make it a better choice for patient A. On the other hand, for patient B, P(D > 3 with superlumab) = 0.60 and P(D > 3 with nivolumab) = 0.35, so the superlumab-versus-nivolumab ARR = 0.25. If superlumab is used, patient B will be much more likely to be alive at three months, thus making superlumab a better choice.

Next, consider another set of preferences for the two patients: Suppose that they both are due to retire in 12 months, and any additional time gained after that would be highly valuable to them and better tolerated even under a therapy with a worse side-effect profile that requires longer and more frequent infusions. For patient A, choosing superlumab over nivolumab would probably add substantially more time to her retirement, thus making superlumab the better choice. On the other hand, patient B would be very unlikely to gain any meaningful amount of additional time for her retirement by choosing superlumab, and thus the QOL advantages of nivolumab make it a better choice.

We can make this treatment choice process explicit by using utility functions to represent and quantify the preferences of clinical outcomes in terms of both survival time and QOL for our patients, where combinations of survival and QOL outcomes having greater numerical utility values are more desirable (Figure 5 and Table 2 and Table 3). The approach of computing expected personal utility functions using probability distributions that include individual patient variables is the most commonly used framework in decision-making research [52,53,54]. We use it here to illustrate how clinicians can choose between treatment options by maximizing the expected utility for their patients. Exponential and isoelastic utility functions are commonly used for medical decisions because they provide simple but plausible representations of risk aversion with respect to survival duration [55]. For the first set of preferences, which focuses on surviving for longer than three months, we use an exponential utility function. For m=mean survival time and QOL= good or poor, the exponential utility function is U_1_(m, good) = (1 − *e*^−m^^⋅0.5^)⋅100, and U_1_(m, poor) = (1 − *e*^−m^^⋅0.5^)⋅85, as shown in Figure 5A and Table 2. For this utility, most gains occur early on, and better QOL is always favored for any given mean survival time (Figure 5 and Table 2). We can use the survival probability distribution shown in Figure 4 to determine the mean survival time, m, for each treatment (n for nivolumab and s for superlumab) and IMDC risk subgroup (f for favorable risk and p for poor risk). For patient A with IMDC-favorable-risk mccRCC, the utility, U_1_, if she is treated with nivolumab, which yields good QOL, will be U_1_(m_n,f_, good QOL) = 100, whereas if she is treated with superlumab, it will be U_1_(m_s,f_, poor QOL) = 85, favoring nivolumab over superlumab. For patient B with IMDC-poor-risk mccRCC, the utility, U_1_, if she is treated with nivolumab will be U_1_(m_n,p_, good QOL) = 76, whereas if she is treated with superlumab, it will be U_1_(m_s,p_, poor QOL) = 80, favoring superlumab over nivolumab (Table 4).

For the second set of preferences, which place more value on QOL differences for shorter survival times, we use an isoelastic utility function, U_2_(m, good) = 10 · m^0.2^/0.2, and U_2_(m, poor) = 9.5 · m^0.3^/0.3, as shown in Figure 5B and Table 3. For the survival probability distribution shown in Figure 4, for patient A with IMDC-favorable-risk mccRCC, if she is treated with nivolumab then U_2_(m_n,f_, good QOL) = 96, whereas if she is treated with superlumab U_2_(m_s,f_, poor QOL) = 104, favoring superlumab over nivolumab. For patient B with IMDC-poor-risk mccRCC, if she is treated with nivolumab U_2_(m_n,p_, good QOL) = 62, whereas if she is treated with superlumab U_2_(m_s,p_, poor QOL) = 54, favoring nivolumab over superlumab (Table 4).

As a formal basis for making personalized treatment decisions, we have advocated assigning numerical utilities to patient outcomes, with the utility assignments tailored to both mean survival time and QOL. This was done to reflect differences in the relative importance of outcomes such as survival time and QOL for patients with different subjective utilities. Our examples show that two different utility functions can lead to different treatment decisions for a given patient. Consequently, the patient’s utility function, as well as their covariates, should guide therapeutic decision making. Although we used exponential and isoelastic utility functions to reflect two qualitatively different types of preferences, other functional forms may be used. We have defined QOL as a binary outcome for simplicity, but a similar approach can be used to generate utility functions for more complex ordinal QOL measures. The numerical utility assignments are subjective in order to reflect risk–benefit trade-offs, which are inherently subjective. This subjectivity is a strength of the methodology, rather than a weakness. If two or more different utility functions are being considered, then the computations can be carried out using each of them, as a method of sensitivity analysis to better inform physicians and patients as to how each utility assignment translates into treatment decisions.

In conventional decision theory, one assigns utilities to combinations of actions and states of nature, which in this case are treatment decisions and parameter values, respectively. The notion of utility has been a longstanding topic in various fields, including economics [56], statistics [52], and game theory [57], but has been underexplored in medicine due to the inherent subjectivity of utilities, as well as difficulties in elicitation. However, as we showed in our hypothetical phase 3 RCT scenario, it is advantageous to include subjective patient preferences explicitly in clinical decision-making. Accordingly, utility-based statistical designs are now increasing being used to facilitate treatment comparisons in RCTs [58,59] and to guide dose finding in phase 1/2 trials [60,61]. The evolving literature thus now includes examples of utility function elicitation from physicians to account for risk–benefit tradeoffs [62,63,64,65]. Within the context of mccRCC, in-depth interviews with clinical experts, patients, and members of the public have been conducted to determine societal preferences and corresponding utility values in the United Kingdom for patients with mccRCC undergoing first-line therapy [66]. Utility functions can also be used to incorporate financial cost considerations for patients and healthcare systems. Such cost–utility analyses traditionally estimate the incremental cost per quality-adjusted life year (QALY) [67,68]. However, as we showed above, different preference-based valuations of health outcomes, such as those emphasizing 3-month or 12-month survival probability, can be used instead of QALYs in cost–utility analyses. Ideally, all pertinent parties should agree on the utility functions incorporating efficacy, adverse events, quality of life, and economic cost. When there are multiple stakeholders with conflicting utilities, a universally optimal decision rule is unattainable. In these situations, dedicated software and graphical tools can be developed to demonstrate the impact of different utility functions used as inputs to generate patient-specific recommendations. Utility elicitation from patients should ideally occur in real-time from the actual patient or family involved, and can start by framing the patient’s hopes within pertinent scenarios, such as “I want to live long enough to see my child graduate from college in 3 months” or “I plan to work for 12 more months and then retire”, which can then be used to generate utility functions such as those shown in Figure 5 and Table 2 and Table 3. Patients can then review the corresponding graphical and tabular utility information, using accessible representation formats, such as those established within the patient decision aid literature [69,70,71], to ensure that the elicited utility functions accurately represent their goals and values.

## 5. Conclusions

Although the estimation of parameters such as HRs is invaluable in carrying out statistical inferences, it is only the first step toward patient-specific decision-making. The next step is to calculate estimates of parameters that correspond to clinically interpretable outcomes informed by individual patient prognostic variables, such as the IMDC score, to account for heterogeneity between patients. Because different outcome scales can produce contradictory inferences, a more complete analysis requires the specification of utility functions that represent each patient’s goals and values, to quantify trade-offs between different clinical outcome variables. The utility functions may be used, along with patient prognostic variables and parameter estimates, to facilitate better-informed patient-specific clinical decisions.

## Figures and Tables

**Figure 1 cancers-13-02741-f001:**
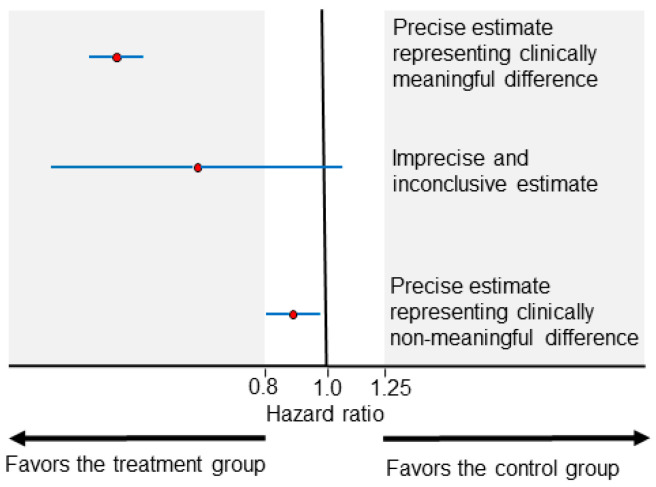
Examples of HR point estimates (red circles) of effect size and their corresponding 0.95 CIs (blue horizontal lines). The areas shaded in gray represent clinically meaningful effect sizes. The top estimate is precise (narrow CI width) and compatible at the 0.05 level with clinically meaningful differences favoring the treatment group. The middle estimate is imprecise (wide CI width) and inconclusive because it is compatible at the 0.05 level with both clinically meaningful and non-meaningful effect sizes. The bottom estimate is precise (narrow CI width) and compatible at the 0.05 level with clinically non-meaningful differences.

**Figure 2 cancers-13-02741-f002:**
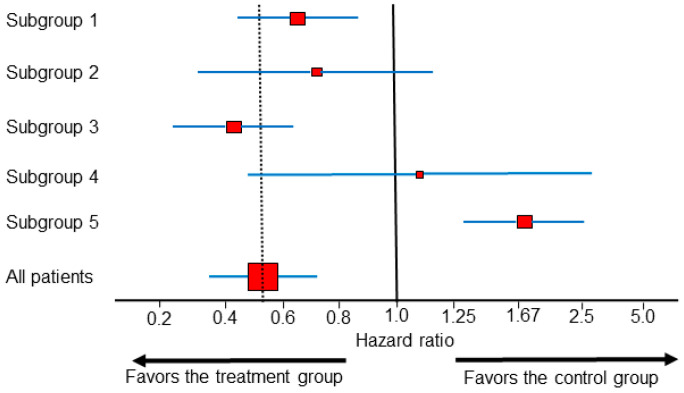
Forest plot with different HR estimates within subgroups from a single randomized controlled trial (RCT). The dotted vertical line shows the overall treatment effect. For each subgroup, the red squares represent the HR estimates, the size of the squares correspond to the sample size, and the blue horizontal lines represent the 0.95 CI. The HR estimates from subgroups 1–4 are statistically compatible with the overall HR effect estimate, and a claim of subgroup difference cannot be made. Subgroup 5 is the only subgroup with a statistical estimate suggesting a difference compared with the overall group.

**Figure 3 cancers-13-02741-f003:**
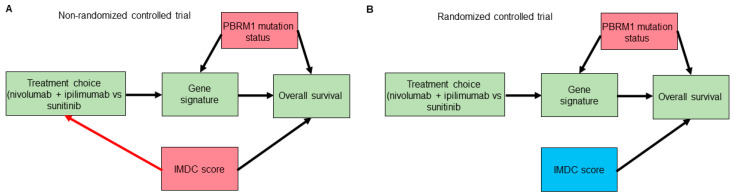
Directed acyclic graphs (DAGs) for a hypothetical trial that identified a gene signature mediating the effect of treatment choice (nivolumab plus ipilimumab or sunitinib) on overall survival. Gene signature is a mediator of the effect of treatment choice on the overall survival outcome. Confounders are highlighted in red boxes. Prognostic variables, highlighted in blue boxes, can improve the power of null hypothesis tests used to evaluate the effect of treatment choice on overall survival. In the non-randomized version of the trial (**A**), the International Metastatic Renal Cell Carcinoma Database Consortium (IMDC) score is a confounder that can bias the effect of treatment choice on overall survival. The red arrow highlights the effect of IMDC score on treatment choice, which is removed through the process of randomization, as shown in (**B**). However, randomization does not remove confounders of the mediator–outcome relationship, such as *PBRM1* mutation status.

**Figure 4 cancers-13-02741-f004:**
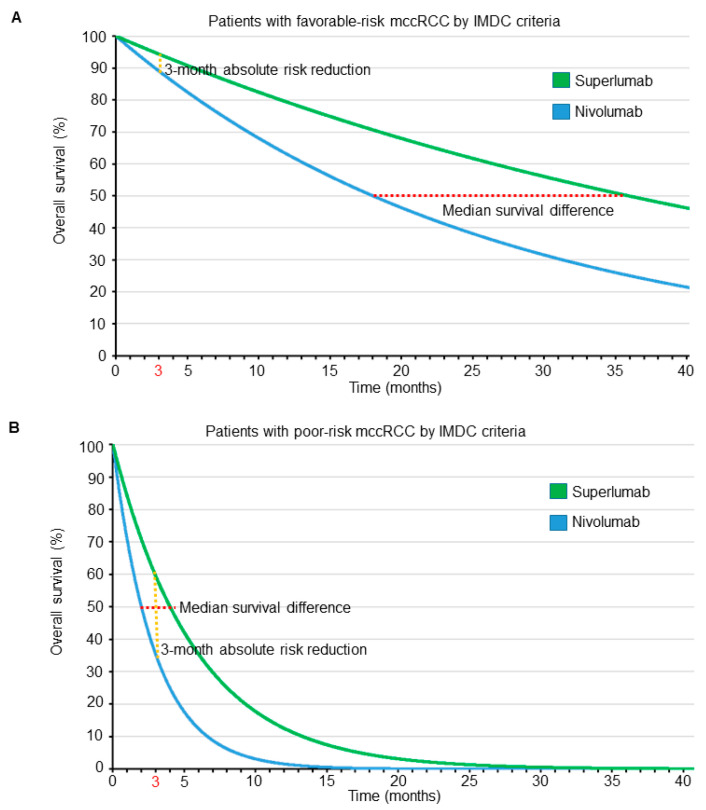
Overall survival time curves, assuming an exponential distribution for patients with metastatic clear cell renal cell carcinoma (mccRCC) treated in a hypothetical randomized controlled trial with either superlumab or nivolumab. The red dotted lines correspond to the median survival difference, whereas the orange dotted lines correspond to the absolute risk reduction at 3 months. The HR is assumed to be 0.5, favoring superlumab over nivolumab, irrespective of International Metastatic Renal Cell Carcinoma Database Consortium (IMDC) prognostic risk classification. For patients with favorable risk determined by IMDC (**A**), the median survival difference favoring superlumab is 18 months, whereas the absolute risk reduction at 3 months is 5%. For patients with poor risk determined by IMDC (**B**), the median survival difference favoring superlumab is 2 months, whereas the absolute risk reduction at 3 months is 25%.

**Figure 5 cancers-13-02741-f005:**
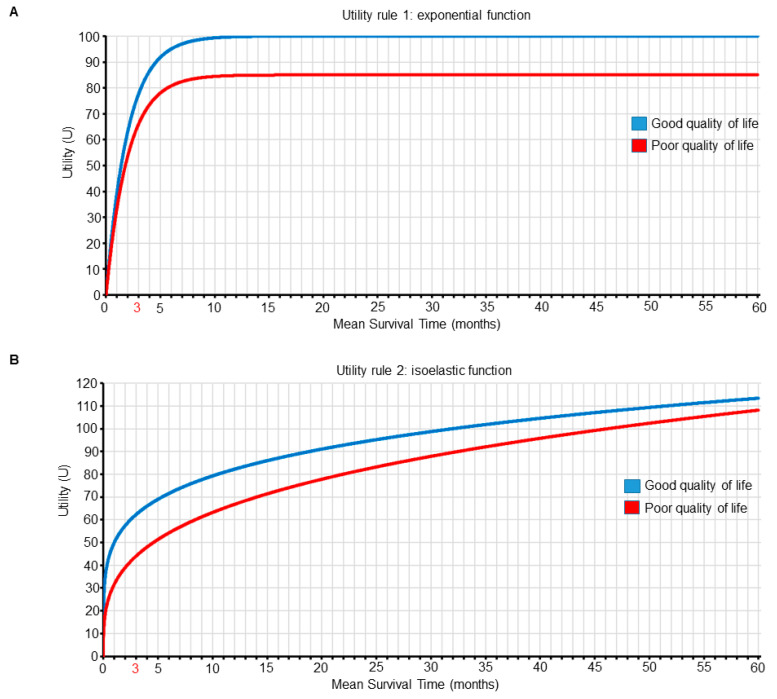
Two different utility functions of mean survival time (m, in months) and quality of life (QOL). Each curve is plotted as a function of m for each QOL subgroup (good versus poor). Panel **A** shows the first utility function, which is an exponential function of m that places more value on surviving for longer than 3 months. The utility function of m for good QOL is U_1_(m, good QOL) = (1 − *e*^−m^^·^^0.5^) · 100, and for poor QOL is U_1_(m, poor QOL) = (1 − *e*^−m^^·0.5^) · 85. Panel **B** shows the second utility function, which places more value on QOL differences for shorter survival times. The utility function for good QOL is U_2_(m, good QOL) = 10 · m^0.2^/0.2, and for poor QOL is U_2_(m, poor QOL) = 9.5 · m^0.3^/0.3.

**Table 1 cancers-13-02741-t001:** Steps needed for patient-specific decision-making.

Task	Scale Used	Example Outputs	Distinct Considerations
Statistical estimation	Estimation scale	Hazard ratios (derived from log_e_ hazards) Odds ratios (derived from log_e_ odds)	Simple, powerful, and flexible summaries of effect size differences; not directly interpretable clinically but can be used to compare clinical effects of interest; non-collapsible parameters may be preferable for categorical and time-to-event models
Clinical outcome prediction	Outcome scale	Median survival, mean survival, three-month survival probability, one-year survival probability, risk difference, absolute risk reduction	Interpretable by clinicians and patients; require knowledge of each patient’s baseline prognostic risk for the outcome of interest; can directly contradict each other depending on which parametric effect is used; collapsible parameters are preferable for categorical and time-to-event outcomes
Clinical decision making	Utility scale	Utility of clinical outcomes	Allows a focus on the clinical outcomes of interest for specific patient prognostic groups; depends on the subjective goals and values of the patient/decision-maker

**Table 2 cancers-13-02741-t002:** First joint utility function, U_1_(m, QOL), of mean survival time (m, in months) and quality of life (QOL) combinations for patients with metastatic clear cell renal cell carcinoma, defined by U_1_(m, good QOL) = (1 − *e*^−m^^·0.5^) · 100 and U_1_(m, poor QOL) = (1 − *e*^−m^^·0.5^) · 85, given that U_1_(0, good or poor QOL) = 0.

	Mean Survival Time (Months)								
QOL	1	2	3	4	12	18	24	30	36
Good	39	63	78	86	100	100	100	100	100
Poor	33	53	66	74	85	85	85	85	85

**Table 3 cancers-13-02741-t003:** Second joint utility function, U_2_(m, QOL), of mean survival time (m, in months) and quality of life (QOL) combinations for patients with metastatic clear cell renal cell carcinoma, defined by U_2_(m, good QOL) = 10 · m^0.2^/0.2 and U_2_(m, poor QOL) = 9.5 · m^0.3^/0.3, given that U_2_(0, good or poor QOL) = 0.

	Mean Survival Time (Months)								
QOL	1	2	3	4	12	18	24	30	36
Good	50	57	62	66	82	89	94	99	103
Poor	32	39	44	48	67	75	82	88	93

**Table 4 cancers-13-02741-t004:** Patient-specific decisions for two patients using data from a hypothetical phase 3 randomized controlled trial that reported a hazard ratio of 0.5, favoring superlumab over nivolumab across all International Metastatic Renal Cell Carcinoma Database Consortium (IMDC) prognostic subgroups.

Parameter	Patient A		Patient B		Conclusions
IMDC prognostic subgroup	Favorable risk		Poor risk		The two patients differ only in their prognostic status
	Nivolumab	Superlumab	Nivolumab	Superlumab	
Median overall survival (months)	18	36	2	4	The median and mean survival differences are more pronounced for patient A compared with patient B
Mean overall survival (months)	26	52	5.8	2.9	
Survival probablity at 3 months	89%	94%	35%	60%	The absolute risk reduction at 3 months is more pronounced for patient B compared with patient A
Utilities based on the first joint utility function (Figure 5A and Table 2)	100	85	76	80	Choose nivolumab for patient A and superlumab for patient B
Utilities based on the second joint utility function (Figure 5B and Table 3)	96	109	62	56	Choose superlumab for patient A and nivolumab for patient B

## Data Availability

Not applicable.

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
