# Peer review of "Making Patient-Specific Treatment Decisions Using Prognostic Variables and Utilities of Clinical Outcomes"

_cancers, 2021, doi:10.3390/cancers13112741_

Round 1
Reviewer 1 Report
Thank you for asking me to review this interesting manuscript on clinical decision making in mRCC. This is an important and overlooked topic in the field of GU cancers.
Overall, this is an excellent overview of statistical modelling and the importance of prognostic factors used in everyday clinical practice. The authors have done an exceptional job explaining complex statistical methods and techniques in a way that many non-experts will understand and appreciate. A very nice review of this topic.
One suggestions would be to include a brief overview of incremental cost-effectiveness ratios in the final section. This may complement the information on patient utility and cost-benefit trade-offs, particularly in the context of a private healthcare system.
Author Response
Comment: “One suggestions would be to include a brief overview of incremental cost-effectiveness ratios in the final section. This may complement the information on patient utility and cost-benefit trade-offs, particularly in the context of a private healthcare system. “
Author response: This is a great point. Utilities can indeed be used to account for cost-effectiveness considerations that are particularly pertinent when making decision at the healthcare system level. We have accordingly now added a discussion of this point on page 13, lines 487-494.
Reviewer 2 Report
Overall I found this paper quite well-written, on a topic of importance in the application of clinical trial data which is quite under-appreciated in practice and very thoughtful in its handling of how we might try to better choose the 'right treatment' for a particular patient given that patient's current status, prognosis and potential outcomes with competing treatment options.
I had some difficulty understanding a few things in Table 2 at first; I think this may just be an awkward flow of manuscript that leaves a few connections unclear to the reader until they have read the whole thing. I see that the authors have provided utility functions in Table 3 and Table 4 as well as Figure 5...it just struck me as odd to encounter "Table 2" first and then wonder what "rule 1" and "rule 2" were referring to & where the utility functions were coming from that computed the numbers in the bottom of the Table. I now assume that these are patient-specific utilities computed for each patient/scenario based on the utility functions in Table 3 and Table 4. If that is correct, I wonder if the authors might consider rearranging somehow to make it more explicit rather than the confused reader first encountering Table 2 and wondering what "rule 1" and "rule 2" are and where these utility functions are coming from. I don't think "Rule 1" and "Rule 2" are ever clearly defined, actually; it's left to the reader to try and piece together that these are connected to the different utility functions.
Regarding the authors' argument that the numerical utility assignments are subjective and that this is a strength, not a weakness, I am generally inclined to agree but have concerns just how easily this can be operationalized for a particular patient's shared decision making process. Is the expectation that each individual patient would define their own utility function, or that they would describe their hopes to the physician (e.g. scenarios as described by the authors, like "I want to work for 12 more months and then retire" or "I want to survive to see my child's wedding") and the physician would try to define utility functions accordingly? Is there potential to build some tools that would facilitate adoption of this thinking into oncology practice? This might be beyond the scope of the paper, but I would like to see a bit more discussion of the potential for implementation.
Author Response
Comment:
“I had some difficulty understanding a few things in Table 2 at first; I think this may just be an awkward flow of manuscript that leaves a few connections unclear to the reader until they have read the whole thing. I see that the authors have provided utility functions in Table 3 and Table 4 as well as Figure 5...it just struck me as odd to encounter "Table 2" first and then wonder what "rule 1" and "rule 2" were referring to & where the utility functions were coming from that computed the numbers in the bottom of the Table. I now assume that these are patient-specific utilities computed for each patient/scenario based on the utility functions in Table 3 and Table 4. If that is correct, I wonder if the authors might consider rearranging somehow to make it more explicit rather than the confused reader first encountering Table 2 and wondering what "rule 1" and "rule 2" are and where these utility functions are coming from. I don't think "Rule 1" and "Rule 2" are ever clearly defined, actually; it's left to the reader to try and piece together that these are connected to the different utility functions.”
Author response: Thank you for bringing this important point to our attention. We have accordingly rearranged this Table (former Table 2; now renumbered to Table 5 in the revised manuscript) to appear after the utility functions have been presented in what are now Tables 2 and 3 (former Tables 3 and 4) and Figure 5. Furthermore, we have revised the Table (Table 5 in the revised manuscript) to explicitly state that what we previously confusingly termed “rule 1” and “rule 2” are indeed the utilities based on the first and second joint utility functions shown in Figure 5 and Tables 2 and 3 of the revised manuscript.
Comment:
“Regarding the authors' argument that the numerical utility assignments are subjective and that this is a strength, not a weakness, I am generally inclined to agree but have concerns just how easily this can be operationalized for a particular patient's shared decision making process. Is the expectation that each individual patient would define their own utility function, or that they would describe their hopes to the physician (e.g. scenarios as described by the authors, like "I want to work for 12 more months and then retire" or "I want to survive to see my child's wedding") and the physician would try to define utility functions accordingly? Is there potential to build some tools that would facilitate adoption of this thinking into oncology practice? This might be beyond the scope of the paper, but I would like to see a bit more discussion of the potential for implementation.”
Author response: This is an important point that indeed merits elaboration. We have now accordingly added a discussion (pages 13-14, lines 482-50) on how the use of utility functions can be operationalized in clinical practice.